# Optimization and characterization of alkaliphilic lipase from a novel *Bacillus cereus* NC7401 strain isolated from diesel fuel polluted soil

Kulsoom Akhter[1]*, Ismat Karim[1], Bushra Aziz[1], Azeem Bibi[1], Jahanzeb Khan[1], Tasleem Akhtar[2]

1 Department of Chemistry, The University of Azad Jammu and Kashmir, Muzaffarabad, Pakistan,
2 Department of Zoology, The University of Azad Jammu and Kashmir, Muzaffarabad, Pakistan

* kalsoom.akhtar@ajku.edu.pk

**Data Availability Statement:** All relevant data are within the paper and its Supporting Information files.

## Abstract

Five *Bacillus cereus* strains including *B. cereus* AVP12, *B. cereus* NC7401, *B. cereus* BDBCO1, *B. cereus* JF70 and *B. specie* JL47 isolated from the diesel fuel polluted soil adhered to the roots of *Tagetes minuta* were screened for lipase production with phenol red agar method. *B. cereus* NC7401 strain successfully expressing and secreting lipase with maximal lipolytic activity was subjected to a submerged fermentation process with five different carbon (starch, glucose, maltose, fructose, and lactose) and five different nitrogen (tryptone, ammonium nitrate, peptone, urea, yeast extract) sources to produce lipase enzyme. Maximum enzyme activity was found with starch (30.6 UmL$^{-1}$), maltose (40 UmL$^{-1}$), and tryptone (38.6 UmL$^{-1}$), and the lipases produced using these sources were named lipase A, B, and C respectively. The total protein content of 8.56, 8.86, and 2.75 µg mL$^{-1}$ were obtained from *B. cereus* NC7401 cultured using starch, maltose, and tryptone respectively. Lipase was stable between temperature range 30–80°C and pH 5–10 whereas optimally active at 55°C and pH 8.0. The enzyme was relatively stable for 10 days at 4°C and its optimum reaction time with the substrate was 30 minutes. It was tolerant to 1.5% (v/v) methanol as an organic solvent, 1.5% (v/v) Triton X-100 as a media additive and 1.5% (w/v) Ni$^{2+}$ as a metal ion. SDS, n-hexane, and Ag$^+$ inhibited lipolytic activity. Oil stains were removed from cotton fabric which showed oil removal efficiency enhancement in the presence of a lipase. Fat hydrolysis of 20, 24, and 30% was achieved following 6 hours of incubation of the fat particles with lipase A, B, and C respectively at a concentration of 20 mg mL$^{-1}$. To as best of our knowledge, this study on lipases extracted from bacteria of Azad Kashmir, Pakistan origin has never been reported before.

## Introduction

Enzymes are utilized as biocatalysts in the industry because they provide many benefits including quick reaction termination, controllable product synthesis, ease of separation from the

**Funding:** The authors received no specific funding for this work.

**Competing interests:** The authors have declared that no competing interests exist.

reaction medium, and flexibility for a wide range of engineering designs [1]. Lipase or triacyl-glycerol lipase (E.C.3.1.3.1), an enzyme of the class serine hydrolase, is a mild and environment-friendly biocatalyst that has become a significant industrial enzyme owing to its importance in the industrial and health sectors. It plays an important role in the digestion of fats by catalyzing the hydrolysis of triacylglycerol (TAG) into mono- and diacylglycerols and free fatty acids [2]. Aside from hydrolysis, lipase is also capable of catalyzing esterification, trans-esterification, alcoholysis, amidation, acidolysis, and aminolysis in a non-aqueous media [3]. Lipase is the most important enzyme due to its broad specificity, high stability, and high stereo, enantio, and regio-selectivity. These characteristics provide this enzyme with great potential at industrial levels such as in food modification, detergents, pharmaceutical, oleo-chemical, leather, textile, cosmetics, perfumery, paper industry, for the production of biofuel, biodiesel, and biopolymers. Moreover, this enzyme is also used in bioremediation and pre-treatment of industrial wastewater [4]. The demand for lipase-based food products is high in the market because of greater hygienic benefits. Other lipase market driving factors include greater consumption of processed foods, dairy, milk & meat products, a rise in changing life-style and dietary habits, and an increase in the number of patients globally suffering from pan-creatic diseases who undergo enzyme replacement therapy. A trend toward greater consumption of meat demands increases in livestock production which results in making lipase application in animal feed, a leading lipase market. The global lipase market was $519.54 Million in 2018 and is expected to reach $715.16 Million by the end of 2023, growing at a CAGR of 6.6% whereas the Asia Pacific has generated the largest revenue in 2014 and is expected to show outstanding growth by the end of the forecast period from 2015–2023. Industrial enzymes are produced primarily through sub-merged fermentation in batch and fed-batch cultures using bacteria. Submerged processes have some advantages over solid-state processes, such as higher homogeneity of the culture medium and more facility to control parameters like temperature and pH [5]. Other factors, such as the type and concentration of nutrients, pH, agitation, and the presence of inducers can affect the productivity of these bio-processes. Research that uses isolated microorganisms from new environments and that uses agro-industrial residues in the composition of media is required to obtain high yields at lower costs. Since the 1980s, several lipases have been identified and purified from different sources, mainly due to recombinant microbial technology, as well as the increasing demand for new and unique properties for these biocatalysts, such as pH, and temperature, stability, and speci-ficity [6]. The microbial origin of lipases in biotechnological and organic chemistry applica-tions is known to be the most used class of enzymes. Microbial enzymes are generated from various classes of microorganisms in industries that include bacteria, fungi, and yeasts. Micro-bial enzymes being economical, produced on large scale, and uninterrupted by seasonal varia-tions with high yield are favored over plant enzymes [7]. Recently bacterial enzymes have gained a lot of attention as potential enzymes in the industries because of their rapid growth rate, ability to survive in different environmental conditions, ability to utilize wide substrates as carbon and nitrogen source, and secretion of different types of extracellular enzymes [8, 9]. The properties of bacterial lipase include its molecular weight, function, and stability under various conditions of pH, temperature, presence of organic solvents, media additives, metal ions, etc. High yield enzyme production has been encouraged through genetic modification in bacterial cells [10]. Bacterial lipases are mainly extracellular and are typically affected by physi-cochemical and nutritional factors. Optimum temperature and pH are both essential for the optimum activity as well as stability of lipases. The most influential criteria to produce lipase are dietary factors, including carbon and nitrogen sources. Bacterial lipases are generally active over a wide pH range (pH 3.0–12.0) [11], while temperature optima ranges from 30–60˚C. Some essential bacterial genera that produce industrial lipases include *Bacillus*, *Pseudomonas*,

and *Acinetobacter. Bacillus sp*. is the major source, producing a variety of soluble extracellular enzymes. These strains are commercially essential industrial extracellular enzyme producers and can be cultured under extreme temperature (40–60°C) and pH (9–11) conditions to produce products that are, in turn, stable in a wide range of harsh environments [12]. Because of the tremendous properties and applications of lipolytic enzymes produced from bacterial strains, especially *Bacillus sp*, it is paramount to identify highly enzymatic bacteria. In the current study, we evaluated the screening of identified *B. cereus* strains isolated from diesel fuel polluted soil adhered to the roots of *Tagetes minuta* to find out the maximum lipolytic activity containing strain, and the effect of different carbon and nitrogen supplements on lipase production through submerged fermentation. We also reported the influence of various factors including pH, temperatures, organic solvents, media additives, and metal ions on the production of lipase by *B. cereus* NC7401 with the objective that it may aid future biotechnological applications of this enzyme. The compatibility of enzymes for commercial application has also been worked out.

## Material and method (*B.cereus* AVP12, *B. cereus* NC7401, *B. cereus* BDBCO1, *B.cereus* JF70 and *B. specie* JL47)

### Screening of maximum lipolytic *B. cereus* strain

Five identified *B. cereus* strains isolated from the diesel fuel polluted soil adhered to the roots of *Tagetes minuta* grown in Muzaffarabad city of Azad Kashmir, Pakistan were screened for maximum lipolytic activity on phenol red agar medium (phenol red, 0.01% w/v; olive oil, 0.1% v/v; CaCl$_2$, 0.1% w/v; agar, 2% w/v and pH 7.3) using well diffusion method [13]. Wells in the plates were filled with bacterial culture in triplicate and the plates were incubated at 37°C for 24 hours. Yellow zones developed around the wells were considered positive for lipolytic activity. The zones of lipolytic activity (mm) were measured and the mean was calculated. The strain showing maximum zones of lipolytic activity was selected for optimization of lipase production.

### Preparation of inoculum and lipase production

The selected bacterial strain culture was prepared in 250 mL Erlenmeyer flasks with 100 mL liquid broth medium (peptone, 5 g/l; beef extract, 3.0 g/l) incubated in a shaker at 37°C and 120 rpm.10 mL overnight culture of bacterial strain was inoculated in each 500 mL Erlenmeyer flask containing 150 mL fermentation media and kept in shaking incubator for 72 hours at 170 rpm and 37°C. The fermentation media was centrifuged for 30 minutes at 6000 rpm, to separate the biomass. Biomass was dried at 50°C until a constant weight was obtained. The supernatant containing protein was used for enzyme assay using olive oil as the substrate. All experiments were done with duplicate flasks with the results reported as the mean of the duplicates. The results were assayed for biomass, lipase activity, specific activity, and protein content.

The medium used for lipase production was optimized in the first design experiment consisting of (% w/v): peptone 0.13; MgSO$_4$.7H$_2$O 0.05; KH$_2$PO$_4$ 0.1; NaNO$_3$ 0.3; yeast extract 0.6 and different carbon sources including starch, glucose, fructose, maltose, and lactose (5 g each) prepared separately in phosphate buffer (pH = 7.0, 100 mM). In the second design experiment with various nitrogen sources, the medium used to maximize lipase production was composed of (% w/v): Peptone 0.13; MgSO$_4$.7H$_2$O 0.05; KH$_2$PO$_4$ 0.1; NaNO$_3$ 0.3, carbon 0.6 and various nitrogen sources including tryptone, ammonium nitrate, yeast extract, peptone, and urea (5 g each) prepared separately in phosphate buffer (pH = 7.0, 100 mM) [7].

The fermentation media showing maximum lipase production were selected for optimization of pH, thermal stability, the effect of the fermentation period, media additives, organic solvents, shelf-life stability, and metal ions by the one-variable at-a-time approach. The effect of pH on lipase production was performed by varying the pH of the culture medium from 3.0 to 10.0 keeping all other process parameters constant. The effect of temperature was determined by varying temperatures from 25–95°C. The effect of the fermentation period on enzyme production was determined in the production medium in submerged fermentation and incubated for 10 days at 37°C and pH 8.0. The sample was withdrawn, and an enzyme assay was carried out. Lipase production media (1 mL) were separately incubated with 1mL of 1.5% media additives solution including SDS, EDTA, glycerol, 2-mercapto ethanol, triton X-100, and tween 80 to study their effects on enzyme production. Effect of ionic sources ($Ni^{2+}$, $Fe^{3+}$, $Cu^{2+}$, $Cr^{3+}$, $Cd^{3+}$, $Pb^{2+}$, $Ag^+$) was performed with various ions at 1.5% level [14].

## Analytical procedures

Lipolytic activity was measured through the titrimetric method with slight modifications [15]. 1 mg enzyme solution was mixed into 1 mL phosphate buffer, added to 3 mL of olive oil, 1 mL of tris-HCl buffer, deionized water (2.5 mL), and incubated at 37°C for 30 minutes. The blank solution containing all other components except the enzyme was also prepared. The reaction was interrupted by the addition of 3 mL ethanol. By adding 2–3 drops of phenolphthalein as an indicator, the released fatty acids were titrated with 0.1 M NaOH. All experiments were repeated three times and the mean values were used. The number of fatty acids liberated in each sample was calculated based on the equivalent of NaOH used to reach the titration endpoint, using the following formula.

$$\mu\text{mol fatty acid/mL sample} = \frac{[(\text{mL NaOH for sample} - \text{mL NaOH for blank}) \text{ x N x1000}]}{\text{volume of reaction mixture (mL)}}$$

μmol fatty acid/mL sample = [(mL NaOH for sample—mL NaOH for blank) x N x1000] ÷ volume of reaction mixture (mL)

Where N is the normality of the NaOH titrant used (0.1 N); 1000 is the conversion factor from miliequivalent to micro equivalent. One unit of enzyme activity is defined as the amount of enzyme required to release 1 μ mol of equivalent fatty acid under standard conditions.

Protein content was estimated by using Bradford's method [14]. Total protein content (TPC) was determined by preparing the reaction mixture containing 3 mL Coomassie blue and enzyme solution (2 mL). The reaction mixture was allowed to stand for 30 minutes, and absorbance was recorded at 595 nm. All the assays were performed in triplicate and the mean value was taken for the estimation of protein concentration. The protein concentration was estimated by plotting the BSA standard curve. Total protein content (TPC) (x) was determined by putting the value of absorbance (y) obtained from the test mixture of enzymes in the straight-line equation (y = 0.099x+0.002) of the BSA standard curve (Table 1).

## Industrial applications of lipase

**Organic solvent stability of *B. cereus* lipase for industrial processes.** Organic solvents including methanol, ethanol, acetone, toluene, and hexane were used to determine the effect of these solvents on lipase activity and stability. Each solvent (1.5% v/v) was incubated with an enzyme produced in starch, maltose, and tryptone media separately for 30 minutes at 37°C and 150 rpm before measuring the lipase activity. To the control sample, the organic solvent was not added under the same experimental protocols [16].

**Table 1. Effect of carbon and nitrogen sources on lipase production by *B. cereus* NC7401.**

| Carbon & Nitrogen sources (5%) | Lipolytic activity (UmL⁻¹) | Specific activity (U/mg) | Protein content(µg mL⁻¹) | Biomass (g/L) |
|---|---|---|---|---|
| Starch | 30.6±0.80 | 3.57 | 8.56 | 25 |
| Glucose | 28.0±0.62 | ND | ND | 21 |
| Fructose | 28.0±0.50 | ND | ND | 19 |
| Maltose | 40.0±0.51 | 4.51 | 8.86 | 28 |
| Lactose | 13.3±0.20 | ND | ND | 10 |
| Tryptone | 38.6±1.50 | 14.03 | 2.75 | 26 |
| Urea | 10.0±0.20 | ND | ND | 4 |
| Yeast extract | - | - | - | - |
| Ammonium nitrate | - | - | - | - |
| Peptone | - | - | - | - |

**Lipase as laundry additive.**   The cleaning ability of lipase on white cotton cloth pieces stained with black grease, chicken blood, and olive oil was evaluated using the reported method [17]. The experiment was performed by preparing different washing performances in Petri dishes to evaluate the effectiveness of lipase as a laundry additive (Fig 1). Cloth pieces were dipped for two hours, washed with tap water, and dried before analyzing the cleaning efficiency of lipase.

**Fat particles hydrolysis ability of lipase.**   Fat particle hydrolysis ability of lipase was checked using the reported procedure [18]. Pieces of beef fat were taken and cut as uniform (0.5 g), dried tiny particles at 30˚C. Different concentrations (10 and 20 mg mL⁻¹) of lipase along with Tris-HCl buffer (100 mM, pH 8.0) were poured into each beef sample and stirred for 3 and 6 hrs at 40˚C in conical flasks separately followed by filtration and drying of the residual mass at 30˚C. A control was also prepared to have the same composition except for the enzyme. The residues were weighed and the average particle mass data were statistically evaluated.

**Statistical analysis.**   Each experiment was repeated three times. The obtained results of optimum conditions were analyzed and compared with standard lipase by one-way ANOVA using SPSS statistics.

## Results and discussion

### Evaluation of lipase activity

Phenol red agar well diffusion method was performed for the screening of lipolytic activity by *B. cereus* strains. Phenol red, or phenol-sulfonphthalein (PSP) 3H-2, 1-Benzoxathiole 1,1,

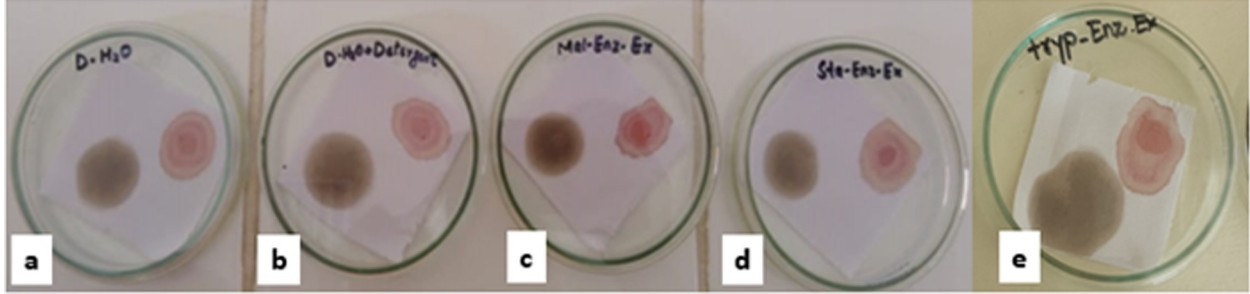

**Fig 1.** Different washing performances to evaluate the effectiveness of lipase as laundry additive (a) 100 mL water; (b) 100 mL water + 1 mL detergent (7 mg mL⁻¹); (c, d and e) 100 mL water + 1 mL detergent (7mg mL⁻¹) + 2 mL solution of Lipase A, B and C respectively.

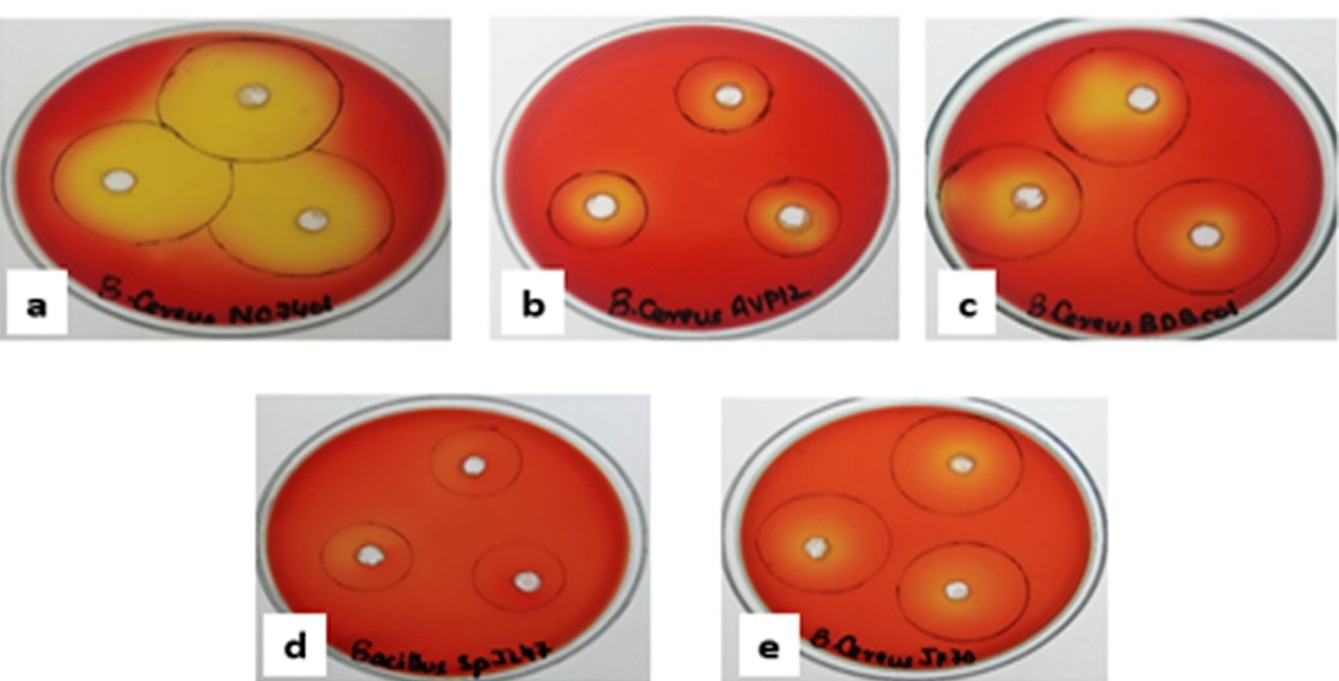

**Fig 2.** Screening of *B. cereus* strains for lipolytic activity with phenol red as pH indicator; a) *B. cereus* NC7401, b) *B.cereus* AVP12, c) *B. cereus* BDBCO1, d) *B. specie* JL47 and e) *B.cereus* JF70.

dioxide a pH indicator dye changes its color from pink (above 7.4) to yellow (below 6.8) [13]. In this method, olive oil was used as a substrate. Lipolytic activity was shown by the appearance of yellow-colored zones around the wells owing to the release of free fatty acid resulting in yellow coloration (Fig 2).

Mean diameter of zones of lipolytic activity (mm) for *B. cereus* AVP12 (1.63), *B. cereus* JF70 (1.43), *B. cereus* BDBCO1 (1.9), and *B. specie* JL47 (1.0) with maximum lipolytic activity provided by *B. cereus* NC7401 (3.8). Based on maximum lipolytic activity *B. cereus* strain NC7401 was selected to produce lipase enzyme through submerged fermentation. Fig 3 shows the scanning electron micrograph (SEM) of *B. cereus* NC7401, which were grown in a nutrient broth medium in a shake flask system after incubating for 24 hours at 37°C and 120 rpm.

## Effect of carbon and nitrogen sources on lipase production and activity

Microbial lipases are primarily extracellular and apart from physicochemical factors such as temperature, pH, and dissolved oxygen, their development is highly influenced by the composition of fermented media. A determining factor in the expression of lipolytic activity has been reported always as a source of carbon and nitrogen because lipase is an inducible enzyme. Sources of lipidic carbon and nitrogen appear to be important in achieving a high lipase yield [6].

In the present study, the production media was supplemented with different carbon sources including starch, glucose, fructose, maltose, lactose, and nitrogen sources such as tryptone, ammonium nitrate, peptone, urea, and yeast. Interestingly, no growth was observed with yeast extract, ammonium nitrate, and peptone. Starch, maltose, and tryptone significantly influenced lipase production from *B. cereus* NC7401, with the activity expressed as $30.6\pm0.80$ UmL$^{-1}$, $40\pm0.51$ UmL$^{-1}$, and $38.6\pm1.50$ UmL$^{-1}$ respectively in a relatively short fermentation period (72 h) (Table 1 and S1 Table in S1 File) when compared with the other carbon and nitrogen

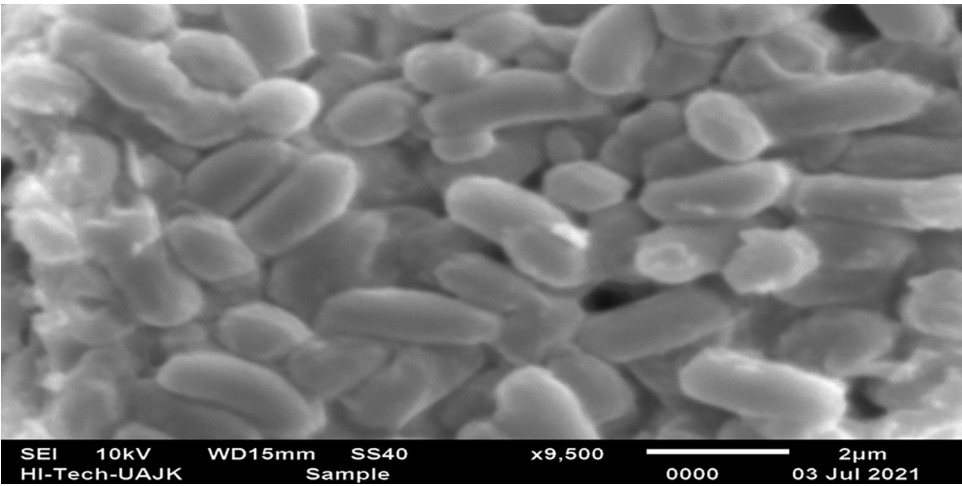

**Fig 3. SEM micrograph of *B. cereus* NC7401, the 24 h old cells that were grown in nutrient broth medium in a shake flasks system.**

sources. Thus, the enzymes produced using starch, maltose, and tryptone were designated as Lipase A, Lipase B, and Lipase C respectively, and were used for the subsequent experiments including characterization and industrial applications. Carbon and nitrogen sources are essential substances to produce energy in micro-organisms, especially in bacteria. The ability of an organism to drive a metabolic reaction and develop in the presence of a particular source of carbon depends on the cell's typical enzymatic machinery [19]. In a microbial system, carbon catabolite regulation (CCR) catabolizes the best sources of carbon supplying carbon and energy most efficiently for growth [20].

Different researchers found various carbon sources such as soybean, corn, sunflower, olive [21], palm and cottonseed oils, wheat bran or soybean bran, triolein, and oleic acid which support the growth and production of lipase [22–25]. Other studies also found that the glucose supplementation to the basal medium acts as an inhibitor of lipase due to the catabolic repression. The effect of various nitrogen sources including yeast extract [26], ammonium salts, sodium nitrate, urea, peptone, and tryptone [27–30] has also been studied previously.

Maltose and yeast extract were reported previously which enhanced lipase production with the lipolytic activity of 30.17 UmL$^{-1}$ and 50.83 UmL$^{-1}$ from *Bacillus cereus*, PCSIR NL-37 [31]. Starch enhanced the production of lipase enzyme from *Serratia rubidaea* with enzyme activity of 15.60±0.20 UmL$^{-1}$ [32] whereas sucrose and fructose enhanced lipolytic activity with 40.67 UmL$^{-1}$ and 31.70 UmL$^{-1}$ respectively from *Bacillus subtilis* PCSIRNL-39 [33]. Lipase production was stimulated in the presence of glucose from *bacillus* FW2 (230 UmL$^{-1}$). *Enterobacter* L7 provided the highest lipase activity using maltose (150 UmL$^{-1}$), glucose (120 UmL$^{-1}$) and lactose (90 UmL$^{-1}$). *Paenibacillus* L2 showed lipase activity of 120 UmL$^{-1}$ using maltose while glucose and lactose provided the same activity of 70 UmL$^{-1}$ [34]. *Bacillus sp*. VITL8 and *Serratia marsecens* showed lipolytic activity of 213.4 UmL$^{-1}$ and 19.2 UmL$^{-1}$ using glucose as a carbon source [14, 35], and in the presence of lactose from *Pseudomonas aeruginosa* JCM5962 (T) (17.236 UmL$^{-1}$) [36].

A combination of 0.6% of tryptone and 0.2% yeast extract was best for enzyme production. Tryptone seemed to play an important role in lipase synthesis, while the yeast extract supported bacterial growth, resulting in higher lipase production from *Bacillus sp*. strain 42 with the activity of 0.174 ± 0.0056 UmL$^{-1}$ [37]. A similar combination was reported for *Bacillus thermoleovorans* ID-1 [38], while 0.3% tryptone and 0.5% yeast extract as nitrogen source was

reported for lipase production by *Pseudomonas aeruginosa* LST-3 [24]. Tryptone supported good growth and lipase production by *P. fluorescens* NS2W (24.5 U mL$^{-1}$) [39]. Peptone was reported to be the best nitrogen source for lipase production from *Serratia marcescens* (28.5 UmL$^{-1}$) [11] and 1% yeast extract for *Bacillus glycinifermentans* MK-840989 (16.8 UmL$^{-1}$) [40]. Yeast extract was also found as the best nitrogen source for lipase activity by *Lysinibacillus* PL33 (400 U mL$^{-1}$) followed by *Lysinibacillus* PL35 (380 U mL$^{-1}$), *Bacillus* L8 (350 U mL$^{-1}$), *Bacillus* FW2 (290 U mL$^{-1}$), *Bacillus* N-PL4 (260 U mL$^{-1}$), and *Paenibacillus* L2 (220 U mL$^{-1}$). Optimal lipase production was obtained by *Enterobacter* L7 and *Paenibacillus* PL2, obtaining the same amount of lipase (290 U mL$^{-1}$), *Enterobacter* PL4 (260 U mL$^{-1}$) and *Stenotrophomonas* N-PL7 (160 U mL$^{-1}$) in the presence of peptone. Casein showed the highest efficiency in *Lysinibacillus* PL35 (270 U mL$^{-1}$), followed by *Bacillus* L8 (240 U mL$^{-1}$), and the same amount of 190 UmL$^{-1}$ for both strains *Bacillus* FW2 and *Enterobacter* L7 [34].

## Effect of pH

The effect of pH on lipase production was inspected by assaying the enzyme activity at a wide range of pH (3–11) using various buffers including citrate phosphate buffer (pH 3.0–5.0), sodium phosphate buffer (6.0–7.0), Tris-HCl (pH 8.0) and glycine-NaOH buffer (pH 9.0–10.0). In an acidic medium, very low lipolytic activity was observed which increased gradually with an increase in pH. Maximum activity was observed at pH 8.0 for the lipase A (32.0±1.1I UmL$^{-1}$), Lipase B (44.0±1.5 UmL$^{-1}$), and Lipase C (40.0±1.60 UmL$^{-1}$) produced by growing *B. cereus* NC7401 in the culture medium containing starch, maltose, and tryptone respectively. At pH 10.0, lipase activity was decreased to 26.6±0.89, 28.0±0.91, and 31.0±0.75 UmL$^{-1}$ for Lipase A, B, and C respectively (Table 2 and S2 Table in S1 File). However, it was still higher than that of acidic medium which indicated that this strain is an alkalophilic bacteria and this lipase is an alkali stable enzyme.

Relatively less activity of 28.5 UmL$^{-1}$ was reported at optimum pH (8.0) for lipase purified from *B. cereus* PCSIR NL-37 grown in a medium containing peptone [23] when compared to our results.

## Effect of temperature

The cultures were incubated at a range of temperature (25–95°C) for 24 hours and the activity of lipase produced was evaluated. Enzyme activity increased with an initial increase in temperature and maximum lipolytic activity of Lipase A (38.0±0.79 UmL$^{-1}$), Lipase B (42.0±0.70 UmL$^{-1}$), and Lipase C (39.0±0.79 UmL$^{-1}$) was achieved at 50°C. However, the activity

**Table 2. Effect of pH on lipase production from *B.cereus* NC7401 cultured using starch (Lipase A), maltose (Lipase B), and tryptone (Lipase C).**

| pH | Enzyme activity UmL$^{-1}$ | | |
|---|---|---|---|
| | Lipase A | Lipase B | Lipase C |
| 3 | 9.34±0.06 | 16.0±0.57 | 18.0±0.65 |
| 4 | 13.30±0.09 | 20.0±0.90 | 22.0±0.85 |
| 5 | 17.30±1.02 | 28.0±1.30 | 29.0±1.00 |
| 6 | 25.30±1.3 | 33.0±1.15 | 33.0±1.11 |
| 7 | 28.0±1.01 | 38.0±1.21 | 37.5±1.3 |
| 8 | 32.0±1.1 | 44.0±1.5 | 40.0±1.60 |
| 9 | 30.0±0.97 | 34.6±0.94 | 36.0±0.56 |
| 10 | 26.6±0.89 | 28.0±0.91 | 31.0±0.75 |

**Table 3. Effect of temperature on lipase production from *B.cereus* NC7401 cultured using starch (Lipase A), maltose (Lipase B), and tryptone (Lipase C).**

| Temperature (˚C) | Enzyme activity UmL$^{-1}$ | | |
|---|---|---|---|
| | Lipase A | Lipase B | Lipase C |
| 25 | 22.0±0.55 | 26.0±0.64 | 24.0±0.68 |
| 37 | 29.0±0.80 | 35.0±0.91 | 32.0±0.56 |
| 50 | 38.0±0.79 | 42.0±0.70 | 39.0±0.79 |
| 65 | 32.0±0.51 | 38.0±0.58 | 35.0±1.0 |
| 80 | 16.0±0.21 | 28.0±0.32 | 29.0±0.33 |
| 95 | 2.67±0.06 | 22.0±0.43 | 15.0±0.05 |

decreased rapidly from 65–95˚C indicating that lipase was stable up to 50˚C (Table 3 and S3 Table in S1 File). These results suggested temperature control is a critical factor for lipase production, as a significant reduction in lipolytic activity resulted from a small variation in the temperature of the system. Thermostable lipases have their strategy depending upon the 3D arrangement of amino acids, to enhance their stability at higher temperatures. These include increasing amino acid close to serine, decreasing helical beta-branched and increasing surface charged residues. For example, it could be an increase in Gly percentage in loops, a decrease in free Cys residues, an increase in polyAla in the lid, and replacement of thermolabile residues with amino acids that show a higher tendency to helix, which enhanced the thermostability of *Bacillus* lipases.

Finally, but significantly, an increase in inverse gamma turn near carboxy and amino end of helixes has been reported to contribute to thermostability [2]. The optimal temperature (45˚C) was reported earlier for lipase from the *Bacillus subtilis* PCSIRNL-39 at pH 7.0 [13]. The majority of the studies with bacterial lipases show a higher activity above 40˚C. Thus, the new lipase from *B. cereus* NC7401 showing high activity and stability in the temperature range of 50–60˚C, may be appropriate for applications in biocatalytic processes.

## Effect of fermentation period

The effect of the fermentation period on lipase activity was examined by assaying the enzyme activity at different incubation times ranging from 2–10 days at pH 8.0 and 37˚C (Table 4 and S4 Table in S1 File). No lipolytic activity was observed up to 24 h of incubation.

It was found to be maximum on the 4th day of incubation with the values of 38.0±0.52, 40.5 ±1.05, and 39.0±0.86 UmL$^{-1}$ for Lipase A, B, and C which decreased slightly till the 10th day. The most significant feature of thermophilic microorganisms is their ability to generate thermostable enzymes with greater operational stability and longer shelf life [41, 42].

**Table 4. Effect of fermentation period on lipase production from *B. cereus* NC7401 cultured using starch (Lipase A), maltose (Lipase B), and tryptone (Lipase C).**

| Shelf life (days) | Enzyme activity UmL$^{-1}$ | | |
|---|---|---|---|
| | Lipase A | Lipase B | Lipase C |
| 2 | 28.0±0.22 | 39.2±0.98 | 32.0±0.24 |
| 4 | 38.0±0.52 | 40.5±1.05 | 39.0±0.86 |
| 6 | 29.0±0.24 | 38.0±0.47 | 29.0±0.71 |
| 8 | 29.5±0.21 | 37.0±0.89 | 27.0±0.42 |
| 10 | 28.0±0.33 | 35.0±0.25 | 25.0±0.36 |

**Table 5. Effect of media additives on lipase production from *B. cereus* NC7401 cultured using starch (Lipase A), maltose (Lipase B), and tryptone (Lipase C).**

| Media additives (1.5% v/v) | Enzyme activity UmL$^{-1}$ | | |
|---|---|---|---|
| | Lipase A | Lipase B | Lipase C |
| Triton X-100 | 34.0±0.9 | 38.0±1.9 | 39.0±1.5 |
| β-mercaptoethanol | 25.9±0.5 | 34.0±1.5 | 32.0±0.98 |
| Tween 80 | 22.36±0.4 | 25.0±0.85 | 24.0±0.6 |
| Glycerol | 22.0±0.34 | 23.0±0.71 | 25.0±0.91 |
| EDTA | 18.8±0.25 | 12.0±0.36 | 15.0±0.23 |
| SDS | 9.0±0.13 | 10.5±0.21 | 13.0±0.21 |
| Control | 33 ±1.00 | | |

## Effect of media additives

The role of various surfactants like Triton X-100, β-mercaptoethanol, Tween 80, glycerol, EDTA, and Sodium dodecylsulphate (SDS) was analyzed on lipase production from *B.cereus* strain NC7401 (Table 5 and S5 Table in S1 File). The control sample was prepared without any surfactant under the same experimental protocol. Maximum lipolytic activity of 34.0±0.9, 38.0 ±1.9, and 39.0±1.5 UmL$^{-1}$ was observed using Triton X-100 while the lipolytic activity of 25.9 ±0.5, 34.0±1.5, and 32.0±0.98 UmL$^{-1}$ was observed using 2-mercaptoethanol for Lipase A, B, and C respectively.

With the nonionic surfactant like triton X-100, an increase in lipase activity might be due to the increased availability of oil substrates in the medium in its presence, or some cases, this detergent itself acts as substrates to increase the enzyme activity. The increase in activity with triton X-100 was also reported for the lipases produced from *B. methylotrophicus* PS3 and *B. cereus* PCSIR NL-37 respectively [23, 43]. Lipase activity was relatively stable in the presence of β-marcaptoethanol. The resistance of the lipase against β-marcaptoethanol might be due to the presence of a lesser number of disulfide bonds exposed on the enzyme surface [16]. Other detergents such as Tween 80, glycerol, EDTA, and SDS decreased lipase production.

Lipases function at the forefront of the hydrophobic and hydrophilic parts, therefore their activity may vary with the surfactant. Naturally, through conformation changes, these surfactants do not deactivate enzymes such as lipase but stabilize it by hydrogen bonding and hydrophobic interactions, and function through an increasing interfacial region of lipid-water as a lipase activator [44]. Ionic surfactants have an adverse effect on lipase activity due to electrostatic interactions. SDS forms a complex with proteins at low concentrations, causing to alter the conformational stability of the enzyme and thus reducing the activity. The interaction of lipase with surfactant is thus dependent upon the type of lipase and surfactant. The enzyme production decreases sharply with EDTA which inhibits enzyme activity through chelation in the active site of the enzyme that alters its tertiary structure, causing it to lose its activity. Depending on the nature of surfactants and their concentration, they can result in either activation or deactivation [16].

## Effect of metal ions

The effect of various metal ions i.e., Ni$^{2+}$, Fe$^{3+}$, Cu$^{2+}$, Cr$^{3+}$, Cd$^{2+}$, Pb$^{2+}$ and Ag$^+$ on lipase production was analyzed and the results were shown in Table 6 and S6 Table in S1 File.

No metal ion was added while all other experimental protocol was the same for the preparation of control. Ni$^{2+}$ has stimulated the lipolytic activity to 37.47±0.52, 37.05±1.1, and 39.5±1.5 UmL$^{-1}$ for Lipase A, B, and C respectively. However, the activity decreased with Fe$^{3+}$, Cu$^{2+}$,

**Table 6. Effect of metal ions on lipase production from _B. cereus_ NC7401 cultured using starch (Lipase A), maltose (Lipase B), and tryptone (Lipase C).**

| Ionic sources (1% w/v) | Enzyme activity UmL$^{-1}$ | | |
|---|---|---|---|
| | Lipase A | Lipase B | Lipase C |
| $Ni^{2+}$ | 37.47±0.52 | 37.05±1.1 | 39.5±1.5 |
| $Fe^{3+}$ | 20.7±0.41 | 23.5.0±0.5 | 31.0±0.56 |
| $Cu^{2+}$ | 14.11±0.22 | 22.36±0.45 | 30.0±0.81 |
| $Cr^{3+}$ | 14.11±0.21 | 20.0±0.41 | 12.0±0.23 |
| $Cd^{3+}$ | 12.94±0.43 | 20.0±0.46 | 16.0±0.78 |
| $Pb^{2+}$ | 11.76±0.19 | 18.82±0.11 | 15.0±0.67 |
| $Ag^{+}$ | 2.35±0.05 | 10.0±0.31 | 9.0±0.24 |
| Control | 33 ±1.00 | | |

$Cr^{3+}$, $Cd^{2+}$, $Pb^{2+}$, and $Ag^{+}$. The involvement of metal ions has a significant role in the function and structure of enzymes. Metal ions may change enzyme conformation resulting in alteration of its activity. They may serve as cofactors in catalysis reactions or form bonds with the amino acid of side chains in the proteins resulting in denaturation of enzyme structure which inhibits enzyme activity [45]. The activity of lipase may also be inhibited by the formation of ionized fatty acid complexes and by altering their solubility and behavior at oil-water interfaces.

$Ni^{+2}$ has also been reported as an activator of lipase from _Bacillus subtilis_ BDG-8 [15] and _Bacillus sp._ VITL8 [46]. Several metal ions ($Mn^{+2}$, $Na^{+}$, $Mg^{+2}$, and $Fe^{+2}$) have been found to cause adverse effects on the activity of lipase, with $Cu^{+2}$ and $Co^{+2}$ having stronger inhibiting effects (less than 5 UmL$^{-1}$) [35]. No clear pattern has been studied in the literature regarding the effect of metal ions on the activity of lipase. The effect is different in different metals, and it may even vary with the same metal if the lipase is from a different source.

## Statistical analysis

The lipolytic activity of newly secreted lipase was compared to the standard enzyme at different optimum conditions by applying one-way ANOVA (results are shown in Fig 4). By applying this statistical software, it is indicated that there is no significant difference in lipolytic activity of standard lipase and secreted lipase. The p values of newly secreted lipase is 0.739 ($R^2$ = 0.157) in tryptone, 0.666 ($R^2$ = 0.250) in maltose, 0.666 ($R^2$ = 0.250) at pH 8.0, 0.872 ($R^2$ = 0.296) at Temperature 30°C, 0.666 ($R^2$ = 0.250) at fermentation period 30 minutes, 0.683 ($R^2$ = 0.380) with Triton X-100, and 0.634 ($R^2$ = 0.295) with $Ni^{+2}$. While at other conditions (without optimum conditions), there is a significant difference that occurs from the standard lipase.

## Industrial applications of partially purified enzymes

**Organic solvent stability of _B. cereus_ NC7401 lipase for industrial processes.** Enzymes must be desirably stable in organic solvents, mainly because of their use in esterification reactions. To study their effects some organic solvents such as hexane, acetone, methanol, ethanol, and toluene were employed in a 1% concentration. To the control sample, the organic solvent was not added under the same experimental protocols. Methanol was found to enhance the activity at 39.41±0.75, 43.5±2.1, and 40±1.8 UmL$^{-1}$ for Lipase A, B, and C respectively while other organic solvents i.e., toluene, acetone, ethanol, and n-hexane reduced the activity (Table 7 and S7 Table in S1 File). The stability of enzymes in organic solvents is an additional property to expand their use in multiple sectors of industry. In aqueous solutions, the majority of enzymes exhibit strong in vitro catalytic activities.

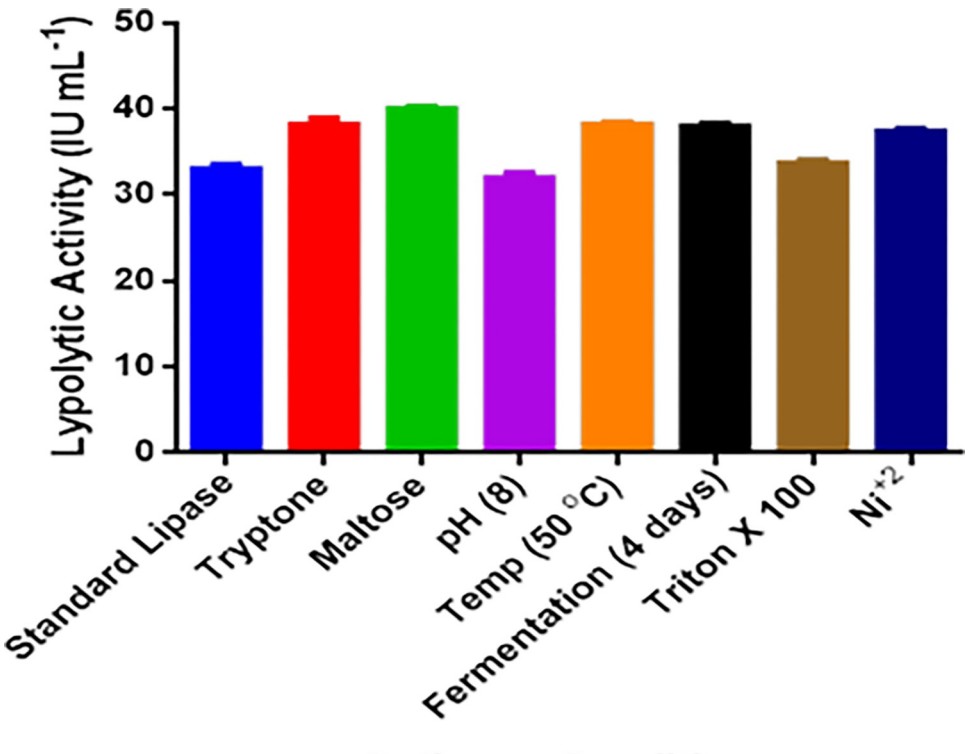

**Fig 4. Statical analysis of different optimum conditions by applying one-way ANOVA.**

Lipases have usually been found to be stable in anhydrous organic solvents, with records of inhibition and enhancement of the activity of enzymes under the influence of such solvents. The stability of enzymes in organic solvents depends on various other parameters such as log P (partition coefficient between aqueous and lipophilic phases) hydrogen binding, polarizability, and functional groups. Lipases are more sensitive and unstable in solvents that are immiscible with water. Few strains of bacteria can survive in butanol, benzene, and toluene [47].

The present study showed high lipase activity in the presence of methanol, whereas the presence of toluene, acetone, ethanol, and n-hexane resulted in reduced enzyme activity. Although water-miscible solvents such as methanol and ethanol have been previously reported to increase lipase activity. For conformational flexibility and enzymatic activity, some water is required to be bound to the surface of the enzyme. The restriction caused by water-immiscible

**Table 7. Effect of organic solvents on the activity of lipase purified from *B.cereus* NC7401 cultured using starch (Lipase A), maltose (Lipase B), and tryptone (Lipase C).**

| Organic Solvents (1.5% v/v) | Enzyme activity UmL$^{-1}$ | | |
|---|---|---|---|
| | Lipase A | Lipase B | Lipase C |
| Methanol | 39.41±0.75 | 43.5±2.1 | 40.0±1.8 |
| Toluene | 14.4±0.51 | 17.6.0±0.5 | 25.0±0.18 |
| Acetone | 10.5±0.12 | 11.6±0.35 | 19.0±0.61 |
| Ethanol | 5.9±0.21 | 12.94±0.21 | 18.0±0.65 |
| n-hexane | 2.35±0.13 | 10.7±0.26 | 15.0±0.33 |
| Control | 33 ±1.00 | | |

**Fig 5.** Cleaning ability of lipase from *B. cereus* NC7401 on pieces of stained cloth; a) Control; Distilled water with stained cloth), b) Detergent with distilled water, c) Distilled water, detergent, and lipase A, d) distilled water, detergent, and lipase B, e) distilled water, detergent, and lipase C.

solvents in the conformation makes enzymes less active in such solvents. The addition of water in water-immiscible solvents to the enzyme suspension significantly boosts enzymatic activity [48]. The enhanced lipolytic activity in methanol was reported for lipase purified from *Bacillus methylotrophicus* PS3 [39]. In contrast, lipase from *Bacillus sp.* was activated by acetone [49].

**Laundry additive.** The purified lipase exhibits stability in the range of 25–55°C, keeping the pH at 7.0, making them candidates for use in laundry detergents. The purified lipase showed a positive response towards grease, olive oil, and bloodstain cleaning within two hours of use. The stains were incompletely removed with water and detergent; however, they were completely cleaned with a mixture of detergent and lipase from samples A, B and C. Fig 5 showed the pieces of cloth washed with the mixture, with enhanced brightness. Lipases from *Geobacillus sp.*, *B. licheniformis*, *B. flexus* XJU-1, and *B. pumilus* SG2 showed higher potential in cleaning the stains.

**Application for fat particles hydrolysis.** Beef fats pose a high risk to the natural environment. The impact of lipase from *B.cereus* strain NC7401 was studied with different concentrations of the enzyme (10–20 mg mL$^{-1}$) on the hydrolysis of fats with incubation times (6 and 24 hours) (Table 8).

A decrease in mass of the fat particles was observed (up to 20%, 24%, and 30%) after its incubation with Lipase A, B, and C for 6.0 hours at maximum enzyme concentration (20 mg mL$^{-1}$). The present results were consistent with *P. aquatica* lipase, exhibiting greater activity against monounsaturated fatty acids, such as linolenic and oleic acids. The results against saturated fatty acids were also consistent with the report [49].

## Conclusion

In conclusion, five strains of *Bacillus cereus* including *B. cereus* AVP12, *B. cereus* NC7401, *B. cereus* BDBCO1, *B.cereus* JF70, and *B. specie* JL47 isolated from the diesel fuel polluted soil were screened with phenol red agar method for lipase production. Interesting findings were

**Table 8. Percent mass reduction of beef fat particle for Lipase A, B, and C.**

| Concentration (mgmL$^{-1}$) | Incubation Time (Hours) | Mass Reduction (%) of beef fat particle | | |
|---|---|---|---|---|
| | | Lipase A | Lipase B | Lipase C |
| 10 | 3 | 6 | 10 | 14 |
| 20 | 3 | 12 | 14 | 22 |
| 10 | 6 | 16 | 18 | 24 |
| 20 | 6 | 20 | 24 | 30 |

observed, where *B.cereus* NC7401 demonstrated the highest lipase activity and was further processed via submerged fermentation using different carbon and nitrogen sources. The optimum activity was obtained at a relatively higher temperature (55°C) and pH (8.0). It was further noticed that the enzyme activity was enhanced in the presence of $Ni^{+2}$, methanol, and triton X-100. These interesting features of bacterial lipase distinguish its bio-catalytic potential for various industrial applications. The current study has laid the groundwork, provided integrated information on some important characteristics of bacterial lipase, and provided a platform for future research on complete purification, optimizing fermentative process parameters, molecular weight, and characterization of said lipase.

## Supporting information

**S1 File.**
(DOCX)

## Acknowledgments

The authors are thankful to the University of Azad Jammu and Kashmir, Muzaffarabad for providing a lab facility to carry out this research work.

## Author Contributions

**Conceptualization:** Kulsoom Akhter, Ismat Karim.

**Data curation:** Ismat Karim.

**Formal analysis:** Kulsoom Akhter, Ismat Karim, Azeem Bibi.

**Investigation:** Ismat Karim, Azeem Bibi.

**Methodology:** Ismat Karim, Azeem Bibi.

**Supervision:** Kulsoom Akhter.

**Writing – original draft:** Kulsoom Akhter, Ismat Karim, Bushra Aziz, Azeem Bibi.

**Writing – review & editing:** Jahanzeb Khan, Tasleem Akhtar.

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
