## [Decision Letter · Decision Letter 0]

5 Apr 2022

PONE-D-22-07934Optimization of Medium for Lipase Production through a novel Bacillus cereus NC7401 Strain Isolated from Diesel Fuel Polluted Soil and its CharacterizationPLOS ONE

Dear Dr. Akhter,

Thank you for submitting your manuscript to PLOS ONE. After careful consideration, we feel that it has merit but does not fully meet PLOS ONE’s publication criteria as it currently stands. Therefore, we invite you to submit a revised version of the manuscript that addresses the points raised during the review process.

ACADEMIC EDITOR: The research work has explained the media optimization and industrial applications of Lipase enzyme from Bacillus cereus NC7401 strain isolated from diesel fuel polluted soil. The manuscript has provided significant data that supports the conclusions and it is well written. However, there are a few suggestions from the reviewers should be consider in the manuscript revision process before it is publication in this journal. 

We look forward to receiving your revised manuscript.

Kind regards,

Dawei Zhang, Ph.D.

Academic Editor

PLOS ONE

Journal Requirements:

"No. The authors did not receive support from any organization for the submitted work. The funders had no role in study design, data collection and analysis, decision to publish, or preparation of the manuscript."

Reviewers' comments:

Reviewer's Responses to Questions

**Comments to the Author**

1. Is the manuscript technically sound, and do the data support the conclusions?

Reviewer #1: Yes

Reviewer #2: Yes

2. Has the statistical analysis been performed appropriately and rigorously? 

Reviewer #1: Yes

Reviewer #2: Yes

3. Have the authors made all data underlying the findings in their manuscript fully available?

Reviewer #1: Yes

Reviewer #2: Yes

4. Is the manuscript presented in an intelligible fashion and written in standard English?

Reviewer #1: Yes

Reviewer #2: Yes

5. Review Comments to the Author

Reviewer #1: In this work, the authors have optimized the lipase production from Bacillus cereus NC7401 strain isolated from diesel fuel polluted soil. In addition, the lipase-producing bacteria were screened through a Phenol red agar well diffusion method. Further, they showed the purified lipase showed a positive response towards grease, olive oil, and bloodstain cleaning. I found that the manuscript has significant novelty and should be considered for publication in this journal. The manuscript is well written, and the authors have provided significant data to support their hypothesis. However, I have a few suggestions for authors to consider during the revision process.

1. The authors should change the title more informatively.

2. In abstract: words merged in many places – check it.

3. The authors should include one more informative keyword.

4. Introduction: The literature review is week. Following references should be included in the manuscript for more readable:

https://doi.org/10.1007/s13205-020-02363-6

https://doi.org/10.1007/s13205-019-1900-8

https://doi.org/10.1007/s13205-016-0481-z

https://doi.org/10.1016/S1872-2067(16)62487-7

https://doi.org/10.1016/j.jiec.2014.12.022

https://doi.org/10.1039/C4RA00066H

5. The authors should italic the genus and species name in the entire manuscript.

6. In result section: the authors should add more discussion with recent references.

7. In many places, the year of the cited publications is missing. Example: Mazhar et al?; Kulkarni and Gadre? The authors should check the citation pattern of the journal.

8. Improve the quality of all the figures.

9. The authors should be uniformed the units and symbols according to the journal format.

10. References must be formatted according to the standard style of PLOS ONE journal. The genus and species name should be italic in entire manuscript.

11. This manuscript has some types of misspellings. It is imperative that these are corrected and the language should be improved.

Reviewer #2: The explained research work was technically good with data that supports the conclusions. All the experiments have been conducted in triplicate with appropriate controls. The conclusions also explained well based on the data presented. All the experimental results are analyzed statistically and significance of results was reported. All the data underlying the findings described in their manuscript fully available without restriction and the manuscript presented in standard English.

The research work explaining the media optimization and industrial applications of Lipase enzyme. The material and method, results and discussion, conclusions are all explained well. The work flow and the parameters used for determining the optimum media constituents for lipase production was excellent and it was helpful for the future researches in optimization of media for different microbial enzymes. I would like to review your coming research work explaining the purification of lipase enzyme in near future.

I would like to inform you to make correction in one place

The topic stating that the Lipase enzyme from bacteria isolated from diesel polluted area, but in material and method you have mentioned like Five identified B. cereus strains isolated from the soil adhered to the roots of Tagetes

minuta grown in Muzaffarabad city of Azad Kashmir.

2nd doubt

After optimization of temperature for the optimum enzyme activity, why did't you used the optimum temperature to determine the incubation time determination?

The application of crude enzyme for removal of dye and meat protein hydrolysis are explained well.

6. PLOS authors have the option to publish the peer review history of their article (what does this mean?). If published, this will include your full peer review and any attached files.

Reviewer #1: **Yes: **Prof. Govarthanan Muthusamy

Reviewer #2: **Yes: **SUDHA S, Assistant Professor, Department of Biotechnology, Sathyabama Institute of Science and Technology

---

## [Author Response · Author response to Decision Letter 0]

30 Jun 2022

Thank you for the opportunity to submit the manuscript revision. As requested, we provide here updates to most of the reviewer's comments with some rebuttals, as well as a detailed description of how we have met/incorporated reviewers suggestions in revised manuscript. Authors really appreciate your keen observations and valuable suggestions to improve the manuscript quality. We have tried to address almost all of reviewers comments and incorporate in the revision attached on desired Journal Template. All the changes are marked red in the text and for each comments, point wise update/detail is as below.

We look forward for acceptance and swift processing of our article. Thanks again for your consideration.

---

## [Decision Letter · Decision Letter 1]

8 Aug 2022

Optimization and Characterization of Alkaliphilic Lipase from a Novel Bacillus cereus NC7401 Strain Isolated from Diesel Fuel Polluted Soil

PONE-D-22-07934R1

Dear Dr. Akhter,

We’re pleased to inform you that your manuscript has been judged scientifically suitable for publication and will be formally accepted for publication once it meets all outstanding technical requirements.

Kind regards,

Dawei Zhang, Ph.D.

Academic Editor

PLOS ONE

Additional Editor Comments (optional):

Reviewers' comments:

Reviewer's Responses to Questions

**Comments to the Author**

1. If the authors have adequately addressed your comments raised in a previous round of review and you feel that this manuscript is now acceptable for publication, you may indicate that here to bypass the “Comments to the Author” section, enter your conflict of interest statement in the “Confidential to Editor” section, and submit your "Accept" recommendation.

Reviewer #1: All comments have been addressed

Reviewer #2: All comments have been addressed

2. Is the manuscript technically sound, and do the data support the conclusions?

Reviewer #1: Yes

Reviewer #2: Yes

3. Has the statistical analysis been performed appropriately and rigorously? 

Reviewer #1: Yes

Reviewer #2: (No Response)

4. Have the authors made all data underlying the findings in their manuscript fully available?

Reviewer #1: Yes

Reviewer #2: Yes

5. Is the manuscript presented in an intelligible fashion and written in standard English?

Reviewer #1: Yes

Reviewer #2: Yes

6. Review Comments to the Author

Reviewer #1: THe authors addressed all the comments properly. Thus, I recommended to accept it in PLOS one. I hope this study will be useful to the scientific community

Reviewer #2: The authors have made all the corrections and brought the manuscript with correct formatting. i could see in many places the

Bacillus species is mentioned as B. specie. I felt instead of B. specie you can mention it as as B.sp.

The authors are justified their results with proper statistical analysis and relevant photos. The references mentioned in the texts are also formatted at the end.

7. PLOS authors have the option to publish the peer review history of their article (what does this mean?). If published, this will include your full peer review and any attached files.

Reviewer #1: No

Reviewer #2: No

---

## [Editor Report · Acceptance letter]

17 Aug 2022

PONE-D-22-07934R1 

Optimization and Characterization of Alkaliphilic Lipase from a Novel *Bacillus cereus* NC7401 Strain Isolated from Diesel Fuel Polluted Soil 

Dear Dr. Akhter:

I'm pleased to inform you that your manuscript has been deemed suitable for publication in PLOS ONE. Congratulations! Your manuscript is now with our production department. 

Kind regards, 

on behalf of

Dr. Dawei Zhang 

Academic Editor

PLOS ONE